# Evaluation of Mobile Applications Usability of Logistics in Life Startups

**Dae-Ho Byun [1], Han-Na Yang [1] and Dong-Seop Chung [2],*** 

[1]   Department of Logistics, Kyungsung University, Suyoung-Ro 309, Nam-Gu, Busan 48434, Korea;
      dhbyun@ks.ac.kr (D.-H.B.); gs1064y@ks.ac.kr (H.-N.Y.)
[2]   Department of Business Administration, Kyungsung University, Suyoung-Ro 309, Nam-Gu,
      Busan 48434, Korea
*    Correspondence: cds7144@ks.ac.kr

**Abstract:** This paper considers the usability of mobile applications operating within a new logistics domain referred to as logistics in life (LIL). The LIL sector has primarily been capitalized on by logistics startups which develop mobile applications or "apps" to provide customized services that penetrate niche spaces outside the reach of traditional logistics firms. The objective of this study is to evaluate whether LIL apps meet usability standards that satisfy users' experiences. As a way to improve usability, problems should be identified through proper measurement and evaluation methods. To derive usability scores, usability testing targeting representative apps from Korea and foreign countries was conducted. The relationship between usability and user interest for each app was determined through big data analytics followed by recommended improvement strategies.

**Keywords:** usability; mobile apps; logistics in life; usability testing; big data analytics

## 1. Introduction

Industry 4.0 has been shown to have positive effects on supply chains and the logistics industry [1,2]. Logistics 4.0, embedded within the broader concept of Industry 4.0, is recognized as one of the fastest growing industrial sectors. As a result of logistics 4.0, the number of logistics in life (LIL) application (app) is growing rapidly. Through O2O (online to offline), omni-channel retail, last-mile delivery, on-demand service, and fresh food chain logistics, logistics app has revolutionized traditional logistics services to enable friendlier, customized services and frequent shipments of smaller quantities to be possible. Recently, the shift from traditional logistics to smart logistics has been led by logistics startups [3]. To aid such a transition, the Korean government established a plan to foster logistics startups that focus on specific areas as a way to foster the development of smart logistics [4]. Startups are commonly defined as Internet-based technology companies that have a high-risk and high-growth potential. The World Bank issued a report that ranked the top startup environments by country. The report listed Hong Kong 5th, Korea 11th, Britain 19th, and followed by the United States at 53rd, Japan at 93rd, and Germany at 114th. In Korea, the most popular startup trend is associated with mobile Internet business opportunities [5].

LIL services, as well introduced in Byun et al. [6], include courier service, fresh food delivery, quick service, matching cargo/luggage owners to carriers, and storage for personal belongings. Customers are able to access these services using mobile apps [7]. Apps are software programs designed to run on smartphones and tablets that allow specific tasks or functions to be undertaken by a user [8]. They are commonly downloaded through application distribution platforms, such as Apple's App Store and Google Play [9]. Mobile information services have revolutionized delivery service methods by facilitating consumer access to information and order placement via apps [10]. Increasing usability

for novice users is a major issue facing companies within the mobile phone environment [11]. In order to acquire and retain customers, as well as ensure profitability, apps must have a user-friendly design that meets the desired goal of the user.

Usability is considered one of the main factors that define the success of an app and refers to methods for improving ease-of-use during the design process. Taking into account the context of use, ISO 9241-11 [12] stated higher usability enables users to achieve their desired goals more effectively, efficiently and with greater satisfactorily. The objective of usability is to achieve quality of use by carrying out tasks in a specific environment, in this case within an app. Usability is defined by five quality components: How easy is it for users to accomplish tasks on their first attempt? How quickly can users perform tasks? How easily can users reach proficiency? How easily can users resolve errors? How pleasurable is the design? [13].

When comparing apps with the Web, users spend more time on apps because of the platform benefits of the App Store and Google Play, which provide fast access speeds and user-friendly interfaces. However, apps require a rigorous design and higher usability, which differs from website usability. As well, apps share the inherent weaknesses of the smartphone, including smaller screens, two operating modes, portrait and landscape, and limited input capabilities, and compatibility with flash scripts [14]. With mobile apps it can take more time to search for content, while scrolling through pages can easily become burdensome, so it is best if apps can avoid these issues. The demand of users to find more data requires better virtualization and the synchronous presentation of search engines on mobile devices [15]. Due to the specific nature of these mobile devices, usability principles applied to the Web cannot be used as they are, as it is more important that mobile devices increase usability beyond Web usability standards [16]. App developers constantly aim to improve their designs to satisfy user satisfaction, a key area in need of continual attention [17]. Highly usable apps positively affect user satisfaction and loyalty, resulting in sustained usage and customer retention. If an app can reduce a consumer's transaction costs or improve the efficiency of transactions, the app is more likely to receive higher satisfaction from customers and broader market acceptance, which in turn leads to greater value retention over time [18]. Kim, Proctor and Salvendy [19] analyzed the positive relationship regarding re-purchases and how usability factors can impact the success of mobile phone product.

This study introduces a new concept of LIL that focuses on bringing greater convenience to people's lives. Starting with food orders and delivery, the scope of LIL services has broadened considerably, and includes fresh food delivery, car washing, car maintenance, on-demand taxi, and courier service. For instance, car washing and car maintenance can be seen as a type of reverse logistics that returns the product (car) to customers after the service has been completed, while on-demand taxis can carry luggage as well as passengers. In addition to transportation, storage, unloading, packaging and information processing functions that represent traditional logistics activities, logistics startups are positioning LIL to create new utility value by reorienting such services through platforms operating through apps. In doing so, LIL has a number of characteristics that distinguish it from traditional logistics. First, it creates new added value in addition to time and place utility. For example, if cargo owners can quickly find an appropriate carrier through an app, customers' convenience is improved. Second, LIL services can be accessed via a free download on Google Play or the App Store, and offer various physical service tasks, such as storing, transporting freight, etc. Third, the primary focus is on addressing issues of convenience in people's lives rather than simply reducing logistical costs. Fourth, it pursues and extends new types of logistics, such as on-demand, convenience and the sharing economy. Fifth, because logistics activities take place mainly within cities, they provide faster delivery times, and can address social problems, such as traffic congestion, as well as environmental and noise pollution.

The success of logistics startups depends on how easily and efficiently users are able to use the app to achieve their desired goal. If the utilization of systems or apps is low, mainly owing to poor overall design problem [20]. To avoid this, it is necessary to evaluate apps from the perspective of usability and in order to formulate guidelines for improvement. The usability of an app enhances user experience

and plays a significant part in the success of the app [21]. In terms of the Web, usability affects user satisfaction which in turn affects intention to use websites [22]. Thus, the evaluation of information systems or an app contributes to customer retention and the sales of additional products [23].

Our interest in this paper is to determine whether the usability of LIL apps is satisfactory to users by discussing how to evaluate them, how to improve their usability, and whether there are differences in the usability of LIL apps in Korea and those from abroad. The primary purpose of this study is to assess the degree to which LIL apps offered by logistics startups are usable. The evaluation of usability includes several activities such as task planning, identifying the evaluation process, and determining source of data. In order to improve usability, problems should be identified and improved upon through accurate measurement and evaluation methods. To evaluate the usability of apps created by Korea's leading logistics startups, as well as apps developed by overseas startups, criteria was derived from a literature review. Usability testing was conducted with subjects in a laboratory setting. Subjects were asked to use the app to find correct answers to given tasks and respond to usability assessment items. Weights were used from the assessment criteria to derive usability scores for each app. After analyzing the evaluation results, improvement guidelines were suggested.

The paper is organized as follows: Section 2 reviews literature on usability evaluation criteria of apps. Section 3 outlines the selection of LIL apps and Section 4 lays out the experiment and the analysis results of the study. Section 5 presents a usability improvement framework and explores trends in big data analytics to assess whether apps that users show interest in score high in usability. Finally, in Section 6 the results of the study are summarized and future points of research are proposed.

## 2. Mobile Usability

There has been abundant research done regarding evaluation criteria of mobile apps and mobile phones. For this paper, evaluation criteria most often citied were used as the initial evaluation criteria for the LIL apps. Kaikkonen et al. [24] stated apps should not be defective in use, be easy to learn, and free of errors. Furthermore, the mobile phone menu should be efficient, effective and easy to use [25]. For the evaluation of mobile phone apps, Ji et al. [26] suggested relevant usability principles include predictability, learnability, consistency, memorability, familiarity, simplicity, feedback, error indication, recoverability, flexibility, effectiveness, efficiency, and effort. Heo et al. [27] highlighted effective interactions between the user and the app as well as highly interactive interfaces. Kim, Jacko and Salvendy [28] argued that usability is related to memory, learning, effectiveness, efficiency, flexibility, and user satisfaction. Among the usability dimensions, more attention should be paid to satisfaction, feedback and efficiency. Nayebi, Desharnais and Abran [29] argued that apps should be easy to use, quickly accomplish tasks, and be user-friendly.

Harrison, Flood and Duce [15] found that effectiveness, efficiency, satisfaction, and errorless were widely recognized as factors affecting usability. Aryana and Clemmensen [30] proposed efficiency, satisfaction, effectiveness, aesthetic, usefulness, simplicity, learnability, understandable, intuitiveness, and attractiveness as usability dimensions. Baharuddin et al. [31] identified effectiveness, efficiency, satisfaction, usefulness, and aesthetics as key usability characteristics of mobile apps. Lai and Zhang [32] argued an app should be easy to use, effective, and satisfying to users. Hoehle and Venkatesh [33] developed Apple's user experience guidelines to conceptualize the usability of apps and develop measurement tools. Hoehle, Aljafari and Venkatesh [21] analyzed Microsoft's mobile usability guidelines and proposed ten constructs representing app usability: graphics, colors, control capabilities, access methods, finger manipulation, suitable font, shape, hierarchy, animation, and screen transitions. Usability requirements should be stated in terms of user performance, satisfaction, acceptability, effectiveness, efficiency, and ease of use [34].

Usability criteria described in 18 studies on mobile phones and mobile apps are shown in Table 1. The five constructs were referenced in more than seven literature articles and used as this study's usability evaluation criteria, as follows: satisfaction, efficiency, effectiveness, learnability, and ease of use. Satisfaction means that the app should provide users with gratification and a positive attitude

towards using it. Satisfaction can be measured by the perception of safety while using the app and the attractiveness of the user interface [35]. Efficiency means the degree to which the app enables a task to be performed in a quick, economical manner, and is usually measured by the time taken for users to complete a task [36]. Effectiveness measures the accuracy and completeness with which users achieve specific goals within a particular context [35]. Effectiveness is a measure of how accurately a user achieves their desired goal [37] and can be computed as a percentage by dividing the number of successfully completed tasks by the total number of tasks undertaken [36].

Learnability measures the task performance of users who have not been previously exposed to the specific app [38]. Novice users should be able to complete tasks quickly with minimum training. Ease of use measures the convenience and simplicity with which the app can be used without exerting a strenuous mental effort. It is a comprehensive concept that involves needing help, error recovery, prompt feedback, and easy-to-understand explanations.

**Table 1.** Breakdown of usability criteria of mobile phones and mobile apps.

| | A | B | C | D | E | F | G | H | I | J | K | L | M | N | O | P | Q | R |
|---|---|---|---|---|---|---|---|---|---|---|---|---|---|---|---|---|---|---|
| Satisfaction | | | | | O | O | | | O | O | O | O | O | O | | | O | O |
| Feedback | | O | | O | O | | | | | | | | | | | O | | |
| Efficiency | | O | O | | O | O | | | O | | O | O | | | | | | O |
| Predictability | | O | | | | | | | | | | | | | | | | |
| Learnability | O | O | | | O | O | | | O | | | | | O | | | O | O |
| Consistency | | O | | | | | | | | | | | | | | | | |
| Memorability | | O | | | O | | | | | | | | | O | | | | |
| Familiarity | | O | | | | | | | | | | | | | | | | |
| Simplicity | | O | | | | | | | O | | | | | | | | | |
| Error indication | | O | | | | | | | | | O | | | | | | | |
| Recoverability | | O | | | | | | | | | | | | | | | | |
| Flexibility | | O | | | O | | | | | | | | | | | | | |
| Effectiveness | | O | O | | O | O | | | O | | O | O | O | | | | | O |
| Effort | | O | | | | | | O | | | | | | | | | | |
| Ease of use | | | O | | | O | O | O | O | | | | O | O | | O | | |
| Aesthetics | | | | | | O | | | O | O | | O | | | O | | | |
| User-friendly | | | | | | | | O | | | | | | | | | | |
| Usefulness | | | | | | | | | O | O | | O | | | | O | | |
| Intuitiveness | | | | | | | | | O | | | | | | | | | |
| Attractiveness | | | | | | | | | O | | | | | | | | | |
| Cognitive load | | | | | | | | | | | O | | | | | | | |
| Accessibility | | | | | | | | | | | | | | | | | | O |
| Interaction | | | | | | | | | | | | | | | | | O | |
| Universality | | | | | | | | | | | | | | | | | | O |

A: [24]; B: [26]; C: [25]; D: [27]; E: [28]; F: [19]; G: [39]; H: [29]; I: [30]; J: [40]; K: [15]; L: [31]; M: [32]; N: [41]; O: [42]; P: [43]; Q: [44]; R: [34].

## 3. Selection of Apps

### 3.1. Selection Principles

The apps to be evaluated were selected after taking into account number of downloads, ratings and reviews among apps in both Google Play and the App Store. As of the first quarter of 2019, app users could choose to download between 2.6 million Android, and 2.2 million iOS apps [45]. A keyword search on Google Play in Korea resulted in 160 freight delivery apps, 50 food or grocery delivery apps, 168 moving service apps, 252 s-hand goods sales apps, 133 flower delivery apps, 63 pet sales or sale of pet supplies apps, and 49 laundry apps.

Although many apps were not available in both stores, the primary consideration was the number of downloads, as the more downloads an app receives, the more popular it can be considered to be. Although ratings can be intentionally manipulated, if the number of reviews is high, the credibility of

the ratings should also be regarded as high. While the number of reviews tends to be proportional to the number of downloads, the credibility of ratings of apps with shorter availability periods should not necessarily be ignored simply due to the small number of reviews.

Considering these factors, a maximum of two apps were selected for each LIL category. The criteria for selecting preferable apps required that the rating score be 3.0 or higher on a five-point scale and the download count should be the highest among similarly-oriented apps. However, categories that did not contain apps with more than 10,000 downloads, such as apps related to moving and packing services, dog sales, and storage for personal belongings, were selected based on highest ratings. The selection criteria for foreign apps required that they be in English and offer services worldwide or within the U.S.

### 3.2. Korean and Forein Apps

The twenty Korean apps are selected as shown in Table 2.

**Table 2.** Koreans apps.

| App Code | Category | App Name | Score (No. of Reviews) | No. of Download |
|---|---|---|---|---|
| D1 | Freight delivery | Okol national freight—freight car service program | 3.6 (2513) | 100,000 |
| D2 | | Sendy—freight service by a delivery van | 4.0 (101) | 10,000 |
| D3 | Moving and packing service | Zimssa | 4.5 (1000) | 100,000 |
| D4 | | Zimcar | 4.4 (507) | 50,000 |
| D5 | Personal storage | Mataju | 4.7 (211) | 10,000 |
| D6 | | Bag Station | 4.8 (25) | 1000 |
| D7 | Food delivery | Baedaltong | 4.4 (60,000) | 5,000,000 |
| D8 | | Mongchon Side Dish | 4.5 (224) | 100,000 |
| D9 | Grocery delivery | Market Kurly-grocery shopping for tomorrow | 3.3 (1000) | 1,000,000 |
| D10 | | Dolsohnese agricultural products | 4.8 (222) | 10,000 |
| D11 | Secondhand goods market | Danggeun Market | 4.5 (30,000) | 5,000,000 |
| D12 | | Hellomarket | 4.5 (45,000) | 1,000,000 |
| D13 | Parcel tracking | 17Track | 4.8 (310,000) | 5,000,000 |
| D14 | | Logi-parcel finder | 4.0 (31,020) | 1,000,000 |
| D15 | Flower delivery | 1644-1644 Flower | 4.8 (7000) | 100,000 |
| D16 | | Baechilsu Flower | 5.0 (165) | 10,000 |
| D17 | Laundry delivery | WashSwat | 4.3 (2000) | 100,000 |
| D18 | | Rewhite | 4.2 (626) | 50,000 |
| D19 | Pet dog sales | Dog ZZang | 4.9 (1000) | 100,000 |

The ten categories of LIL are as follows: finding cargo trucks, estimating moving cost and finding moving companies, finding locations to store luggage, ordering food and groceries, buying and selling of second-hand goods, tracking parcels, delivering laundry, and buying and selling pets. As shown in Table 3, the following categories of apps target U.S. or international users, excluding South Korean, and are available in English.

**Table 3.** Foreign apps.

| App Code | Category | App Name | Score (No. of Reviews) | No. of Download |
|---|---|---|---|---|
| F1 | Freight delivery | DAT | 4.4 (1396) | 100,000 |
| F2 | | Shipa Freight | 4.3 (84) | 10,000 |
| F3 | Personal storage | BAGBNB | 4.4 (1324) | 50,000 |
| F4 | | StoreMe | 4.0 (46) | 5000 |
| F5 | Food delivery | Glovo | 4.0 (217,000) | 10,000,000 |
| F6 | | Deliveroo | 4.3 (110,000) | 5,000,000 |
| F7 | Grocery delivery | Honestbee | 3.2 (10,000) | 1,000,000 |
| F8 | Secondhand goods market | Shpock | 4.0 (390,000) | 10,000,000 |
| F9 | | Trovit | 4.0 (56,000) | 5,000,000 |
| F10 | Parcel tracking | Package Tracker Express | 4.1 (3117) | 100,000 |
| F11 | | Parcels-Track packages from Aliexpress, eBay | 4.0 (9738) | 500,000 |
| F12 | Flower delivery | LolaFlora | 4.0 (1600) | 100,000 |
| F13 | Laundry delivery | Laundrapp | 3.5 (1026) | 100,000 |
| F14 | | Laundryheap | 4.0 (255) | 10,000 |

## 3.3. Categories of Logistics in Life

As a result of Logistics 4.0, various types of new logistics services have emerged within the smart logistics field, such as apps catering to LIL domains. The company DHL included convenience logistics, on-demand logistics, anticipatory logistics and urban logistics within the broader smart logistics sector [46]. On-demand logistics is logistics in which demand determines supply by immediately securing supply as customer demand arises or orders are placed. In addition, LIL combines with O2O to adopt a business model that connects offline stores with platforms via apps. Examples include ordering and delivering food using an app without needing to visit a restaurant directly, finding a truck to transport luggage or packages, and collecting and delivering laundry without having to visit the laundry mat directly by utilizing reverse and forward logistics. The apps in such categories include D1, D2, D7–D10, D17, D18, F14, and F15.

City logistics can enhance customer service by providing an optimal delivery solution to complex traffic conditions within cities [47]. Eventually, the objectives of LIL apps should be to focus on improving the quality of life of people by offering solutions to traffic congestion, ensuring traffic safety, and reducing energy consumption [48,49]. Fair and responsible logistics aims to prioritize social responsibility and sustainability within logistics companies. Solutions include reducing packaging boxes, using eco-friendly packaging materials, cutting carbon dioxide emissions by preventing empty vehicles from returning after deliveries, processing or recycling waste in an environmentally-friendly manner, and parcel tracking technology which ensures transportation reliability. Such apps include D11, D12, F9, and F10 that allow the buying and selling of second-hand items, and D13, D14, F11, and F12 apps that allow real-time tracking of a parcel's delivery status.

Fresh chain logistics transports refrigerated and frozen foods that must be sold quickly to customers within 24 h by using cold chain technology to maintain a constant temperature during transport [50]. To maintain freshness, last-mile deliveries are shipped to customers' doors and on-demand orders are fulfilled immediately. Apps in this area include D9, D10, F7, and F8.

## 4. Research Design

### *4.1. Usability Testing*

According to mobile human computer interaction research methods [51,52], various usability evaluation methods are being developed to assess and improve usability of interactive systems. In addition, many studies regarding usability testing have already been conducted on mobile apps [53,54]. One such method, the heuristic evaluation proposed by Nielsen [55] is a cost-effective inspection method as a result of its speed and affordability, but it is limited in that skilled professionals may not reflect real users, and therefore make it difficult to achieve an accurate understanding of usability. For this reason, field tests and laboratory tests are recommended in order for real users to perform a given task within a testing environment, as there was no significant difference in the number of usability problems that occur in the laboratory versus filed test [24]. However, laboratory testing is recommended over field testing since it enables a more focused experiment to occur within a controlled environment and more convenient than field testing. For usability testing, it is widely assumed that as few as 5 to 10 participants are needed per testing period using an iterative development process. Such testing is capable of finding 99% of the usability problems [56].

Figure 1 shows a research flow including research questions. For this study, usability testing was conducted in a quiet room over a five hour period, including breaks. Participants were given a stopwatch and listened to the testing procedure. We made it possible for them to download one or two apps and look around. The participants were college-aged students majoring in logistics with experience using logistics apps. A total of 28 people participated in the test, with 17 Android and 11 iPhone users respectively. Usability assessment can produce relatively accurate results, even with a small group of 4–5 participants, yet the results are subject to changeability depending on the choice of evaluators [39,55]. It is recommended that the number of participants in an ordinary test be 5 to 10 participants per testing round [24].

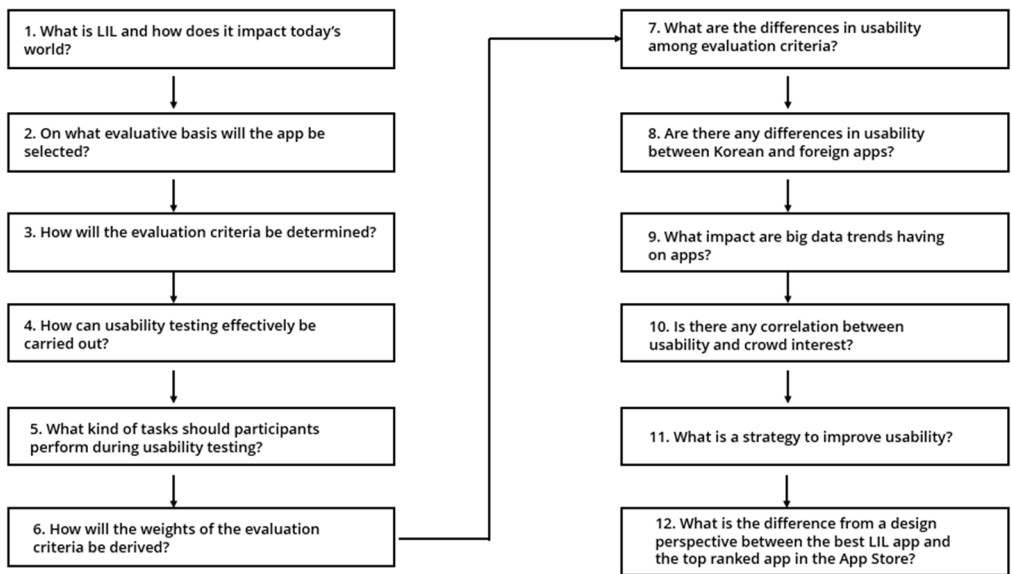

**Figure 1.** Research questions.

The participants were asked to find the correct answers to specific questions and assess the app, irrespective of finding the answers within the five minute period. After completing each task, the participants evaluated their level of satisfaction on a seven-point scale for a given usability assessment item. The first task was to download and install the app, the second task was to find the correct answers to questions by navigating through the app, and the third task was to proceed to the order stage until payment was requested. The subjects were compensated for their time and effort.

*4.2. Question Items*

Question details for each app are shown in Table 4. Two questions were given for each app, and the answers to the question could be found by simply navigating the app's first or second page. The answers could be obtained through simple navigation of the app.

**Table 4.** Question items.

| App Code | Question | App Code | Question |
|---|---|---|---|
| D1 | (1) Is Chuncheon city listed as the address of the loading place? (2) What is the company email address? | D18 | (1) Can I pay with credit card points? (2) Can I sign up with a KakaoTalk account? |
| D2 | (1) Is it possible to transport moving parts? (2) Can I specify a time other than the start date of transportation? | D19 | (1) What is the minimum sale price? (2) Is the blue-white variety on sale? |
| D3 | (1) How many carriers will arrive up to 24 h after getting a quote? (2) Can I move a large massage chair? | F1 | (1) How many digits should I enter when I sign up? (2) Can I inquire about the freight rates immediately in real time? |
| D4 | (1) How many tons can I move? (2) What's the price of a box for moving to a studio apartment? | F2 | (1) How many digits or more should the password be for a membership? (2) Is door-to-door transportation available? |
| D5 | (1) Can I get insurance on my stowage? (2) How much does it cost to keep two bicycles for a month? | F3 | (1) How many storage locations are there in Busan? (2) What time can belongings be kept at Victoria Station in London, England until? |
| D6 | (1) What is the cost of storing one bag per day? (2) What is the name of the storage facility located in Nampo-dong? | F4 | (1) What is the charge for a small bag for one hour? (2) Can you use the map to find the nearest storage location from your current location? |
| D7 | (1) What is the price of a DoubleX2 burger at the Lotteria shop at Kyungsung University? (2) Can I order flowers? | F5 | (1) Can I log in with my Kakao Talk account? (2) Does it have a Spanish language setting? |
| D8 | (1) If I order early morning delivery, what time can I get it at the earliest? (2) What's the customer service number? | F6 | (1) When I enter a restaurant called "Chilango," how many minutes do I need to wait for delivery? (2) Can customers who order food without delivery pick it up themselves? |
| D9 | (1) How much is the chicken breast? (2) Can I order without a membership? | F7 | (1) Can you deliver it in China? (2) What's the price of two liters of the "Meiji Pasteurized Fresh Milk" sold in Singapore? |
| D10 | (1) Is the customer center open on weekends and holidays? (2) What is the price of the fresh potatoes? | F8 | (1) Is there a pet category? (2) How many kinds of foreign currency can the product price be marked? |
| D11 | (1) Can I buy and sell a used car? (2) Can I chat without a membership? | F9 | (1) If you select a country as a state, will the vehicle type be detected? (2) What dollar amount is the lowest price that is searchable? |
| D12 | (1) Do they have golfing items in the product categories? (2) How much can I pay using credit card points? | F10 | (1) Is there a Korea Post in the personalized carrier list? (2) What is the URL of the website you should visit to create an account? |
| D13 | (1) How many invoice numbers can I register? (2) Is there a 'mother' & 'kids' category? | F11 | (1) Can I scan the barcode to use when entering the tracking number? (2) Is there a DEXI on the carrier list? |
| D14 | (1) Can I get an estimated cost for the move? (2) How many delivery points can I get when I look up OK Cash-bag points? | F12 | (1) The price of '19 Rose Red Hearts' is $30.15, but does it include a vase? (2) If I'm going to search for a product for a gift. How many types can be chosen from? |
| D15 | (1) What is the domestic phone number? (2) What is the most expensive congratulatory wreath members can get? | F13 | (1) If I'm going to search for a product for a gift. How many types can be chosen from? (2) Is the price of the product marked in dollars? |
| D16 | (1) What's the lowest price among Asian flowers? (2) What is the lowest price among rubber trees? | F14 | (1) Is the laundry delivery charge marked? (2) How much is the service price for 15 pounds of laundry? |
| D17 | (1) What points will be set aside for the first order? (2) How many hours can I set for laundry? | | |

*4.3. Weight of the Evaluation Criteria*

The usability assessment criteria for the apps were derived from a literature review after searching a number of academic articles over the last ten years using the keywords, "mobile application + usability criteria", in two academic databases (i.e., Emerald Engineering, Computing and Technology

Collection, and ScienceDirect). Emerald includes 230 journals published by Emerald Publishing and ScienceDirect has 1013 journals published by Elsevier Science.

Of the evaluation criteria mentioned in Table 1, five criteria were selected based on how often they were referred to in various literature articles and include: satisfaction, efficiency, effectiveness, ease of use, and learnability. The more references to each criterion in the existing literature, the more importance they can be assumed to have. Therefore, the weights were set in proportion to the sum of the number of keywords mentioned in the two academic databases, and is shown in Table 5. The efficiency criterion was the most heavily weighted, while the satisfaction criterion was given the least weighted.

**Table 5.** Weights of usability criteria.

| | Emerald | | ScienceDirect | | Total | |
|---|---|---|---|---|---|---|
| **Criteria** | **No.** | **Rank** | **No.** | **Rank** | **No.** | **Weight** |
| Satisfaction | 5611 | 5 | 11,964 | 5 | 17575 | 0.06 |
| Efficiency | 5665 | 4 | 95,577 | 1 | 101,242 | 0.37 |
| Effectiveness | 6001 | 3 | 43,621 | 3 | 49,622 | 0.18 |
| Ease of use | 8907 | 2 | 70,939 | 2 | 79,846 | 0.29 |
| Learnability | 9020 | 1 | 18,158 | 4 | 27,178 | 0.10 |

After performing the task, participants answered a questionnaire on the 5 criteria using a seven-point Likert rating, ranging from "strongly disagree" to "strongly agree". The following measurement questions were developed for the basis of each criterion: Were you satisfied with the apps?; Were you able to quickly find the right answer to the question?; Were you able to find many right answers to the question?; Was the app easy to use?; Was it easy to learn how to use the app? The overall usability scores of the evaluation criteria by the test participants are shown as Table 6.

**Table 6.** Usability score for criteria.

| Criteria | Score |
|---|---|
| Satisfaction | 60.9 |
| Efficiency | 52.3 |
| Effectiveness | 60.3 |
| Ease of use | 61.8 |
| Learnability | 64.3 |
| Average | 59.6 |

*4.4. Measurement of the Effectiveness and Efficiency*

Effectiveness is an indicator of whether the participant can find a correct answer to a given question. Effectiveness can be measured using the completion rate of tasks [38]. It is computed to an average score by giving 100 points if the correct answer is found and 0 points if not found. For example, if the correct answer for the first question is found, but the second one is not, then the effectiveness is scored as 50 points. Efficiency is an indicator of how quickly the answer can be found. A stopwatch was used to measure the time taken for the participants to find the correct answers to each question. Efficiency is used as a tool to measure the time it takes to finish a task [38]. In order to relieve participants' time pressure, we explained to them beforehand not feel pressured and prevented them from checking the time during the test, and instead notified them of the end of the test using an alarm. The efficiency of a question was given zero points if the selected answer was incorrect or the correct answer was not found within five minutes of starting the question. The overall efficiency of each app became the average efficiency of the two questions.

## 5. Results

### 5.1. Demographic Survey

The ratio of mobile phone operating systems resulted in more Androids than iPhones being used. The survey found that 71.5 percent of the participants use apps more than three hours a day, while 89.3 percent use more than two apps a day as shown in Table 7.

**Table 7.** Demographic survey.

| Item | Category | Ratio (%) |
|---|---|---|
| Operating system | Android | 60.7 |
| | iPhone | 39.3 |
| App's daily hours of use | <3 h | 28.5 |
| | ≥3 h | 71.5 |
| App's daily numbers of use | <2 apps | 10.7 |
| | ≥2 apps | 89.3 |

### 5.2. Usability Score

As shown in Table 6, the overall average evaluation criteria score of 59.6 was not high when the seven-point scale for each evaluation criterion was converted into 100 points. The criterion with the highest score was learnability and the lowest criterion score was efficiency. The weighted average was 57.6 when the weights of the evaluation criteria were multiplied by the responses of each participant. This weighted score was lower than the average score of 59.6 when the weights were not applied. There was no criteria for how many points or more a usability score was required to satisfy, but generally there will be less of a need to improve usability with a score of 80 or above.

ANOVA was performed to analyze differences in mean values between each evaluation criteria. The null hypothesis was that the population mean of the five evaluation criteria was equal, at a significant level of 5%, Probability (F > F critical) = 0.05, F > reject-value, then the null hypothesis was rejected. Since F = 62.7 > F critical = 2.42, we can conclude that the mean value of the evaluation criteria is significantly different.

The score and ranking of usability for each app is shown in Table 8. The app with the highest usability reflecting the weighted criteria was D9, followed by D8, D11. The apps that scored lowest on usability were F6, F1, and D17 apps. The study found that there were more highly usable Korean apps than foreign apps, particularly regarding the Korean food delivery apps.

Similarly, ANOVA was conducted to analyze differences in the mean value of each app's usability score. Assuming that the 33 apps have an equal population mean, F = 1.00 < F critical = 1.53 at the significant level of 5%, the null hypothesis was accepted, and can be concluded that the usability scores of the apps are equal. First, an F-test was performed to determine whether the variance of the two populations was equal. Next, a T-test was performed to compare whether there was a difference in the usability scores of Korean apps and foreign apps. The mean value of Korean apps was 65.6 compared to 52.0 for the foreign apps. In the F-test, because the *p*-value of 0.377 was greater than the significant level of 5%, the null hypothesis was accepted, and it can be concluded that the variances of both groups were equal. Assuming equal variances of the two groups, because *p*-value = 0.6006 > 0.05, the null hypothesis was accepted and the means of the two population was equal.

The user interface of operating systems can affect user experience. In an empirical study, iOS possessed a favorable brand image, while Android was noted for its functional performance [57]. For this study, a test was conducted to determine whether there was a difference between the usability scores of iOS and Android. The results of the T-test for usability differences between the iOS user group and Android user group showed that the *p*-value was 0.862, larger than 0.05. The null hypothesis

was accepted and it can be concluded that there was no difference in the mean value between the two groups.

**Table 8.** Usability score for each app.

| App Code | Category | Score | Rank | App Code | Category | Score | Rank |
|---|---|---|---|---|---|---|---|
| D1 | Freight delivery | 52.4 | 20 | F1 | Freight delivery | 23.7 | 32 |
| D2 | | 64.7 | 11 | F2 | | 47.4 | 25 |
| D3 | Moving and packing service | 50.7 | 21 | F3 | Personal storage | 52.9 | 29 |
| D4 | | 64.6 | 12 | F4 | | 47.1 | 26 |
| D5 | Personal storage | 55.7 | 16 | F5 | Food delivery | 43.8 | 30 |
| D6 | | 70.7 | 8 | F6 | | 18.4 | 33 |
| D7 | Food delivery | 69.1 | 9 | F7 | Grocery delivery | 46.1 | 28 |
| D8 | | 87.9 | 2 | F8 | Secondhand goods market | 76.3 | 5 |
| D9 | Grocery delivery | 93.8 | 1 | F9 | | 66.6 | 10 |
| D10 | | 63.0 | 14 | F10 | Parcel tracking | 49.2 | 24 |
| D11 | Secondhand goods market | 82.7 | 3 | F11 | | 73.5 | 6 |
| D12 | | 50.4 | 22 | F12 | Flower delivery | 53.8 | 18 |
| D13 | Parcel tracking | 61.9 | 15 | F13 | Laundry delivery | 64.5 | 13 |
| D14 | | 45.5 | 29 | F14 | | 49.7 | 23 |
| D15 | Flower delivery | 78.9 | 4 | | | | |
| D16 | | 46.9 | 27 | | | | |
| D17 | Laundry delivery | 23.8 | 31 | | | | |
| D18 | | 55.0 | 17 | | | | |
| D19 | Pet dog sales | 71.3 | 7 | | | | |

*5.3. Big Data Analytics*

Naver is the largest portal site in Korea and operates a database that provides various indicators related to Korea through search terms. Naver's Data Lab shows how often keywords and sub keywords were searched for on personal computers and mobile devices, representing them on a 100-point scale. An analysis of big data trends was done to determine the monthly average scores for the Korean apps. After searching 2019 trends using app names and app service areas as keywords the highest monthly average was D3 at 80.2 points. Apps with a monthly average of 28 points or more included D3, D4, D9, D11, D16, and D14, and are displayed in Figure 2.

Real world activities with relatively higher frequencies of occurrence showed higher interest trend data, such as moving and packing services, secondhand goods sales, and fresh food deliveries. However, interest in food delivery, in which many apps are widely available was low, as well as interest in laundry delivery and agricultural products, which have recently penetrated niche markets. In regards to foreign apps, Google Trends was used to assess interest levels. The app with the highest average score was F3 with 61.1, followed by F5 with 54.0, F11 with 52.3, and F2 with 47.2, and is shown in Figure 3. The apps that trended poorly were F13 with 9.9, followed by F1 with 18.3. However, unlike the Korean apps, the foreign apps showed a higher interest in regard to personal storage and food delivery. The usability and interest of each app are shown in Figure 4. The correlation coefficient between the two groups was 0.06 and therefore it can be concluded that usability and interest have no relationship.

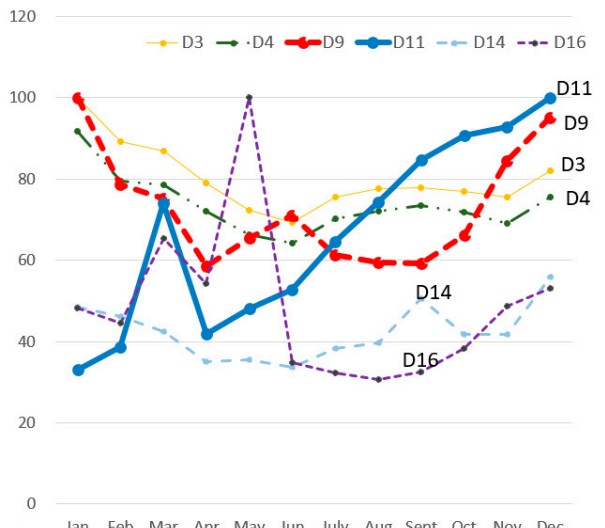

**Figure 2.** Big data trend for Korean apps.

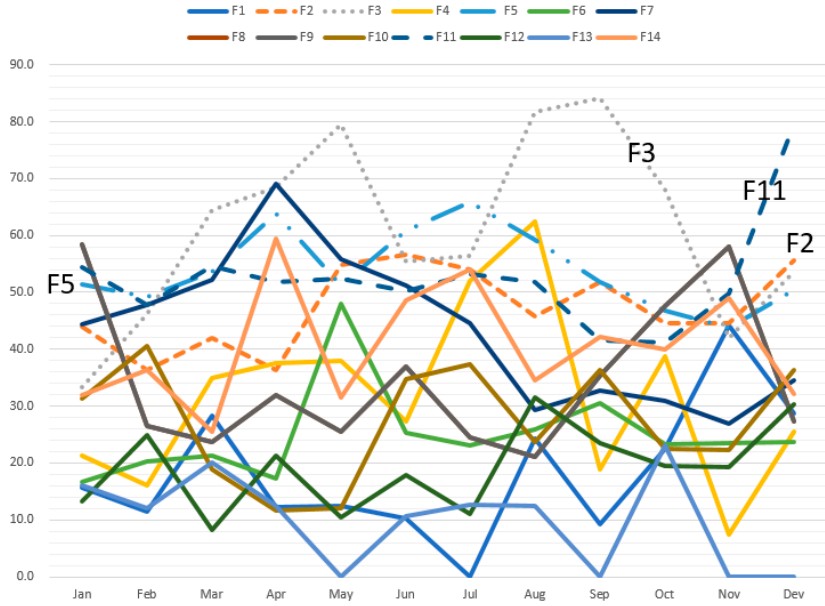

**Figure 3.** Big data trend for foreign apps.

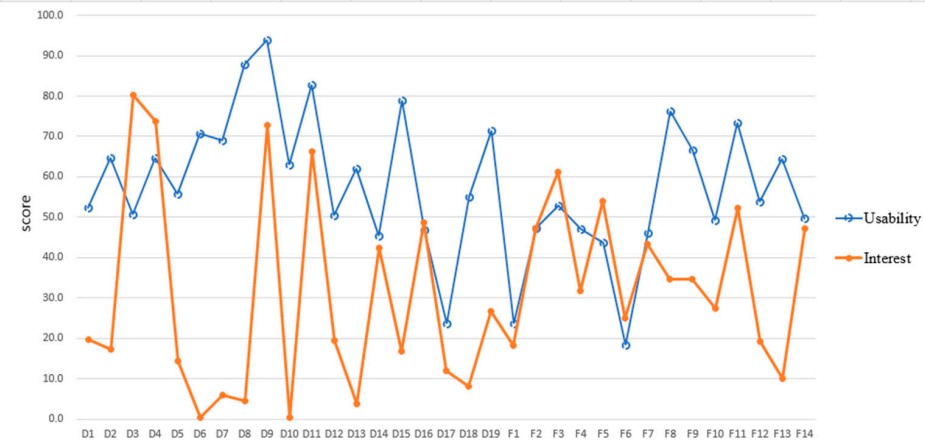

**Figure 4.** Trend of usability and interest.

### 5.4. Usability Improvement Strategy

In order to classify apps that require usability improvements, two quadrants were analyzed, with the *x*-axis serving as the degree of usability and the *y*-axis as the degree of interest. The value of interest of the big data trends is shown in Figures 1 and 2 and the usability scores are shown in Table 8. Assuming that 50 points out of 100 is the average, each point can be represented by the coordinate value of the quadrant. Quadrant I can be classified as high in interest and usability, quadrant II shows high interest but low usability, quadrant IIII denotes low interest and usability, and quadrant IV marks low interest, but high usability.

The improvement strategy for quadrant I is to maintain and reinforce the status quo, quadrant II to improve focus, quadrant III to gradually improve, and quadrant IV to continually maintain overall performance. As such, the first area in need of improvements to usability is in quadrant II. Typically, the higher the interest, the higher the importance, yet when an app with a high level of interest possesses less usability, there is an immediate need to focus on improving its usability.

The usability-interest analysis for apps is shown in Figure 5. Since some of the apps are located in quadrants I, the need for improvement is low, indicating the need to enhance the status quo and keeping up the app's usability. For apps in quadrant III, usability and interest need to be gradually improved.

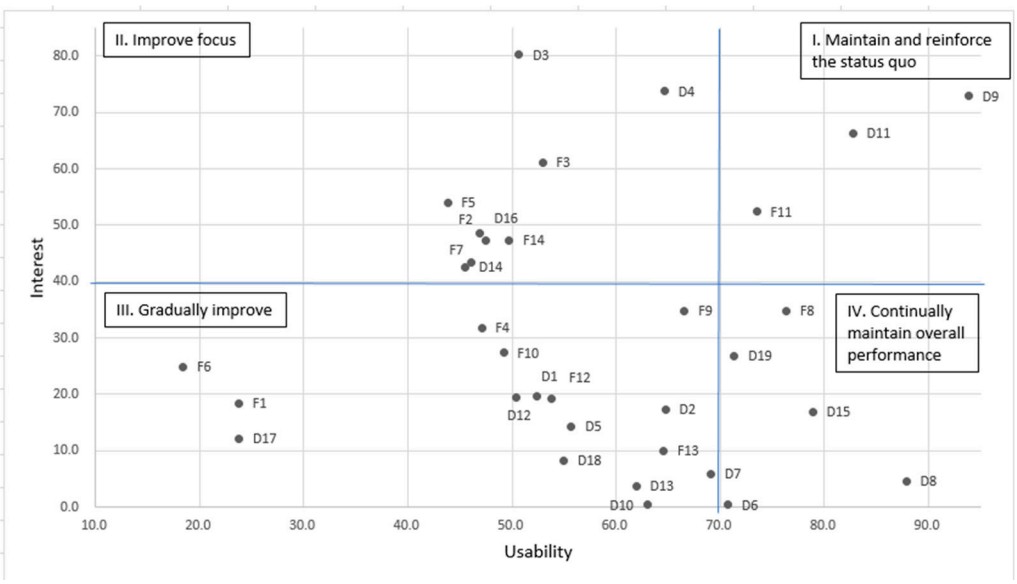

**Figure 5.** Usability-interest analysis for apps.

Similarly, in order to determine usability improvement strategies, usability-importance criteria was analyzed as shown in Figure 6. Importance serves as the weight for the evaluation criteria, and usability is the average score evaluated for the entire app, as shown in Tables 5 and 6, respectively. The usability score for each criterion was above 50 points for all apps, but there was no criterion that scored greater than 50 on importance. In other words, all of the evaluation criteria were found to be located in quadrant IV and therefore performance needs to be sustained continually. In particular, efficiency needs to be improved upon first because of its high importance but low usability scores.

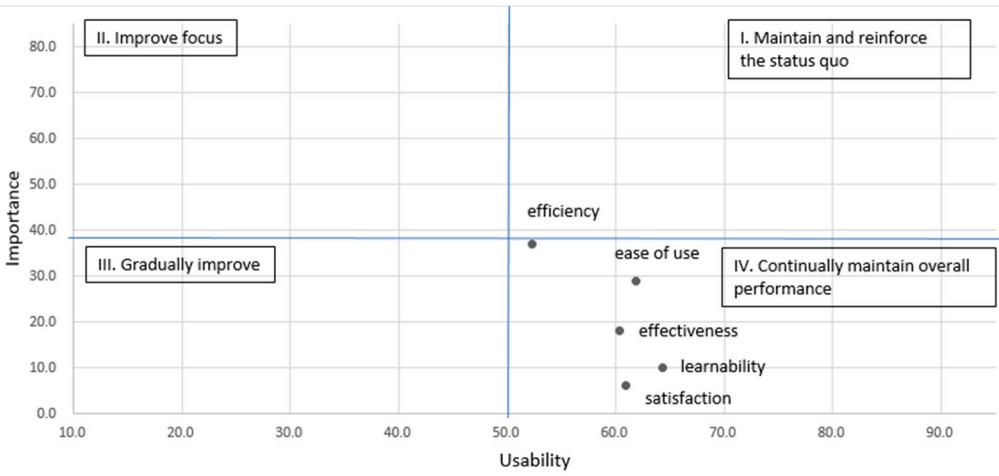

**Figure 6.** Usability-importance analysis for criteria.

## 6. Conclusions

Logistics startups mainly develop apps and strive to enhance customer convenience, thereby improving people's quality of life. In order for a LIL startup to succeed, it must build a highly usable app that encourages customer retention and purchases. The usability testing results showed Korean apps to have higher usability than the foreign apps. However, the overall usability score for Korean and foreign apps was below 60 points on a 100-point scale, suggesting further improvements are still required. By logistics sub-categories, the Korean grocery delivery app scored highest in usability, while the foreign freight delivery and food delivery apps scored lowest in usability. Although the efficiency criterion was found to be the most important from the literature review, the actual test results showed that learnability had the highest usability score. Among the two user groups, there was no difference in usability between the iPhone and Android groups. Big data analytics showed that usability and interest have no relationship. As a result of classifying the applications according to their level of interest and usability as a strategy for improving usability, there were 27% apps that needed to be greatly improved, with 24% apps needing their performance to be continually sustained, and while 49% apps need gradual improvements to be made.

As a result of logistics 4.0, the development of apps continues to grow. However, increased competition between apps allows uses to easily switch between different apps, potentially uninstalling them without ever having used them. Increased usability contributes to increased sales through returning customers. To accomplish this, it is necessary to evaluate apps from the perspective of usability and formulate guidelines for improvements as apps serve highly important platforms for customers to interact with businesses. This paper is significance in that Korean LIL apps using an experimental method called usability testing and proposed strategies to improve usability. LIL apps pursue eco-friendly sustainable distribution method, while contributing to the growth of the sharing economy, and improve the quality of life for current and future generations to come. Therefore the results derived from this study may be used as a guideline for the government's policy regarding the promotion of logistics startups.

The limitation of this study include not being able to assess all types of LIL businesses, while the apps selected do not represent the degree of usability for all apps. However, the selected apps have many users and are representative of the LIL category. Therefore, the reliability of the results is considered high for the study's purpose. Future research should aim to design a prototypical user interface to reflect usability improvements which can then be used to address issues in other similar apps.

**Author Contributions:** Conceptualization, D.-H.B. and H.-N.Y.; methodology, D.-H.B.; analysis, D.-H.B.; writing-original draft preparation, D.-H.B.; data curation, D.-H.B.; writing-review and editing, H.-N.Y.; funding acquisition, D.-H.B. and D.-S.C. All authors have read and agreed to the published version of the manuscript.

**Funding:** This work was supported by Jungseok Logistics Foundation Grant.

**Conflicts of Interest:** The authors declare no conflict interest.

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
