# Peer review of "Evaluation of Mobile Applications Usability of Logistics in Life Startups"

_sustainability, doi:10.3390/su12219023_

Round 1

Reviewer 1 Report

This manuscript evaluates logistics in life (LIL) apps for usability.  LIL apps typically have main goals including customer convenience, efficiency for user, and ease of user to become fluent in the apps’ features.  The study includes Korean LILs as well as top ranked apps independent of national origin; and utilizes Korean subjects to evaluate Korean and non-Korean apps.  Outcomes of the work are input and guidance to government policy for logistics companies. 

Section 3.1

The statement about the number of reviews being high is ambiguous.  How many reviews is high?  And conversely the statement about small number of reviews underestimated is also ambiguous.  How does one know which apps with low numbers of reviews has accurate reviewer scores?  This is not scientifically sound and is fundamental to the study.

Section 3.3

Paragraph 1:  This section refers to the apps as “categories” while the tables and the rest of the text refers to “sectors.”  To keep the reader in sync, this section should use the same terminology as in the table, namely “sectors.”

Paragraph 2:  This manuscript is being submitted to the journal “Sustainability”.  The paragraph from lines 196-205 is the one paragraph in the entire manuscript that touches on sustainability.  And 8 of the 33 apps surveyed pertain to sustainability.  But sustainability is not the focus of this work.  The main focus is usability of mobile apps for consumer logistics, and specifically LIL, not the services that the company provides to society, not the economic benefits, not the impact to the environment, and therefore not with a focus on sustainability.  This manuscript would be better suited to a journal focused on logistics and computer applications in logistics, not “Sustainability”. 

Section 4.1

Line 212 - “According to mobile HCI research methods” needs to be defined and a reference added. 

Line 225 – “expertise” needs to be defined.  That word choice indicates that testing was completed to define expertise and therefore the method to define expertise should be clearly specified.  Otherwise, perhaps “familiarity” or “experience” might be a better term to use.  Actually, the choice of pre-selecting people with higher than average skills in this field would skew the usability scores which introduces bias into the study. 

Lines 230-233  vs Line 235 – There seems to be confusion with the word “tasks” in two different contexts.  Within 5 lines, there’s discussion about 3 tasks and then 2 tasks.  Table 4 is titled “Task items” and two questions are given for each app.  Perhaps “2 tasks” in line 235 should say “2 questions”.  As written, using only the word task for both contexts, will likely cause reader confusion.

Section 4.3

Line 246 – This statement is broad and seems overreached.  It depends on how well the literature review was done.   Only 2 academic databases were searched.  Also would be interesting if the categories have changed in importance over the 10 years – essentially the lifetime of iPhones.

Section 4.4

Line 257-259 – How aware were subject of the stop watch?  If subjects know they are being timed, the results can be skewed due to some people being flustered and others working at a more rapid pace than normal to “beat the clock”.  People behave differently under time pressure and that detail can skew the experiment. 

Line 258 – The efficiency measure combines effectiveness and efficiency, so these are no longer independent factors.  Now only 4 independent factors remain.  So the point about 5 independent factors in this study is wrong.  C2 and C3 are one.

Table 5 – Due to methodology, Science Direct is heavily dominant.  So, the weighting is essentially Science Direct alone.  This means that the literature review is really only based on one academic database not two.   Perhaps it would be better to use percentages of each criteria rather than count so that Emerald (with its lower numbers) would contribute to the weighting.  Learnability and efficiency are the important criteria for this particular scenario.  It has the potential to down-play learnability and up-play efficiency. 

Table 6 – To ensure that your small sample is relevant, does this data match the ratio in South Korea?  If this work is meant to guide policy, which affects all citizens, using demographic survey of students specializing in logistics seems to introduce bias.  How applicable are the results of this work to the larger population?

Line 273-4 and Table 7 – The weighting resulted in a change from 57.6 to 59.6 which didn’t substantially alter the score result.  The weighting didn’t change the conclusion for the average, but changes to variance might have been affected.  That isn’t included in the data, so we don’t know.  With learnability being the highest score and efficiency being the lowest score, the weighting of .1 and .37 respectively significantly alters the results.  Given the earlier discussion about Table 5, this could result in a fundamental flaw to the study and dramatically different results be obtained. 

Line 276 – The comment about being “less need to improve usability with a score above 80…” seems a reach.  On what basis?  Why?  Is this common knowledge?  If so, need to cite why 80 is ok.  If not from a citation, need to justify this choice. 

Section 5 – Distribution values need to be added to summarize values in Table 8.  And that needs to be done before the statistical test results are stated.  With the flaws in setting up the experiment, the statistical analysis cannot be validated. 

Minor points/suggestions for clarity:

Line 38 – “resolves around” should be “revolves around”

Line 153 – Add a count in the final column of Table 1 which helps the reader to see which ones have the highest references.

Line 219-221 – Suggestion to break this into two sentences. “Since the results of field and laboratory tests have shown no significant differences, either method is acceptable for validity.  Laboratory testing is recommended over field…” 

Author Response

Oct.20, 2020

Dear, Reviewer

Sustainability

We are greatly thankful to the learned and knowledgeable reviewers for appreciating our efforts in the present work.

First of all we would like to thank you for your constructive criticism on the manuscript, which has definitely helped in improving its quality in the revised version and also more importantly our research direction.

The manuscript has been thoroughly revised by considering all the points raised by the reviewers. Especially the revised parts in the text are marked in sky-blue letters for easy understanding.

We hope our manuscript is suitable for publishing in the Economic and Business Aspects of Sustainability.

We are looking forward to hearing a good news. Best regards

Sincerely,

Dong-Seop Chung, Ph.D

Department of Business Administration, Kyungsung University, Suyoung-Ro 309, Nam-Gu, Busan 48434, South Korea; Tel: +82-51-663-4437(Off); Mobile: +82-10-4221-7144 FAX: 82-51-627-6830 Email: [email protected]

Reviewer 2 Report

The topic of the paper is interesting and up-to-day because the development of all areas in logistics is under the pressure of external factors. In the paper, I found several errors:

There are missing sources 34-39 between L135 and L141, where are used sources 33 and 40. Also, references 34 and 35 are not located anywhere in the text. The description of the research methods, which authors used, is not clear.
From the complex point of view, the paper seems to be average paper with a standard level of processing. It includes a lack of methodology description, no stated hypothesis (authors verify it at L278-281), no description of the used method for evaluation of gained values. Also, there is necessary to extend the results' description in chapter 5.3, 5.4, and 5.5.

Author Response

(The authors gave the same response as above.)

Reviewer 3 Report

Thank you for the fresh, informative and well-written research. It is interesting to see how Korean and American startup companies in logistics sector with novel business models (tied to the use of mobile applications) perform. The finding of usability somewhat correlating with popularity of an app, was especially insightful. Please find comments and improvement recommendations on your manuscript in the below:

  • Introduction section refers Windows Phone Store. which has already been closed due to competition and exit of Microsoft from phone business. Please do not mention it. Do also check Blackberry's store name - it has changed (also due to competition).
  • Would suggest authors to add research problem statement through research question or questions in e.g. Section 1.
  • Check the used language, typos, etc. (e.g., rows 113, 126, 138, 173, 473)
  • In Table 4 following sentences does not make that much sense: "If I'm want to register the product I want to sell. How many currencies can you get a quote in?" Please rewrite.
  • Do connect used framework properly on information systems evaluation research (check e.g. ERP system evaluation: Häkkinen et al. (2008) Life after ERP implementation – Long-term development of user perceptions of system success in an after-sales environment. Journal of Enterprise Information Management, 21:3, pp. 285-309.)
  • Table 2 and 3: It is recommendable to not merge cells in the “Sector” column in order to avoid confusion
  • Section 4.3: Since the research at hand is novel, it might be valuable to justify the used criteria with something else in addition to the previous literature. What if, for example, criterion “User-friendly” is particularly important in studying Logistics in Life applications performance? Also, basing the weights of different criteria on existing literature might produce results that are applicable to the previous studies concentrating on different types of applications, but is it relevant here?
  • Section 4.4: You describe how effectiveness and efficiency were measured. Satisfaction was measured with a survey. How were the other criteria measured (Ease of use and Learnability)? In addition, effectiveness is reasonable to measure the way you have described, but efficiency of an app could be better measured as how promptly a generic use case can be completed (e.g., just ordering a basic meal, no additional questions asked).
  • Section 5.4: Quadrant IV applications have well-adjusted usability but no interest towards them. Could it be recommended for these applications to focus in raising interest (e.g., via marketing)?
  • Section 5.5: It is insightful to benchmark the best performing studied application with a popular one such as Netflix. However, it should be more justified why Netflix (media streaming service) was chosen, since the studied application (D9) is used for grocery delivery. These applications have vastly different purposes, use cases and target audience. Some of the aspects where D9 performs better can be explained simply with the differences in purpose of the application. For example, registration and payment methods need to be more diverse for grocery deliveries, since those deliveries attract one-time purchasers. However, Netflix uses business model where all the content is available via registration and monthly payments. It is difficult to develop a similar business model for grocery deliveries. Maybe D9 should be compared to the global leading grocery delivery application (or something in highly similar field, e.g., fast-food delivery app, like Postmates, sold recently to Ober in billion dollar deal)?
  • Conclusions (Section 6): Row 490-491, it is claimed that Netflix is the top app in App Store. Earlier in section 5.5., it was stated that it is 5th in the ranking. Also, it would enhance the scientific value of this manuscript to reflect the conclusions to the earlier studies mentioned in the literature review. How did the results compare to the previous results? Were the studied theories possibly reinforced or contradicted?
  • Conclusions needs citations on earlier research, and then clearly illustrating what is your key contribution in here, and in what parts your research supports earlier research findings.

Author Response

Oct.20, 2020

Dear, Professor

Sustainability

We are greatly thankful to the learned and knowledgeable reviewers for appreciating our efforts in the present work.

First of all we would like to thank you for your constructive criticism on the manuscript, which has definitely helped in improving its quality in the revised version and also more importantly our research direction.

The manuscript has been thoroughly revised by considering all the points raised by the reviewers. Especially the revised parts in the text are marked in sky-blue letters for easy understanding.

We hope our manuscript is suitable for publishing in the Economic and Business Aspects of Sustainability.

We are looking forward to hearing a good news. Best regards

Sincerely,

Dong-Seop Chung, Ph.D

Department of Business Administration, Kyungsung University, Suyoung-Ro 309, Nam-Gu, Busan 48434, South Korea; Tel: +82-51-663-4437(Off); Mobile: +82-10-4221-7144 FAX: 82-51-627-6830 Email: [email protected]

Reviewer 4 Report

Please refer to the review file.

Author Response

(The authors gave the same response as above.)

Round 2

Reviewer 1 Report

Edits have improved the paper.  Responses to the comments and clarification added to the paper improve the readability and understandability of this work to a broader population of readers.

Author Response

Oct.28, 2020

Dear, Reviewer

Sustainability

We are very grateful for to continued sincere comments and thoughtful advice of reviews. The manuscript has been thoroughly revised by considering all the points raised by you. Especially the revised parts in the text are marked in sky-blue letters for easy understanding.  We tried to refine the English to improve the readability of the sentences.

We are looking forward to hearing a good news. Best regards

Sincerely,

Dong-Seop Chung, Ph.D

Reviewer 2 Report

Authors correct defined errors, which were mentioned in the first review round. At this moment, the paper reaches a standard level of scientific papers.

Author Response

(The authors gave the same response as above.)

Reviewer 3 Report

Still do not understand inclusion of Netflix as benchmark for logistics app companies. Also provided further justification to make this comparison is not valid. Authors can do better job in here. In my opinion Netflix comparison should be removed and replaced as advised in the first review report.

In conclusions section can not find references to earlier research findings - these are missing, although authors indicated that situation could be opposite.

Author Response

(The authors gave the same response as above.)
